# Electrical Characteristics of 3D Trench Electrode Germanium Detector with Nested Complementary Cathodes

**DOI:** 10.3390/mi14112051

**Published:** 2023-11-01

**Authors:** Mingyang Wang, Zheng Li, Bo Xiong, Yongguang Xiao

**Affiliations:** 1School of Materials Science and Engineering, Xiangtan University, Xiangtan 411105, China; 202131550147@smail.xtu.edu.cn (M.W.); boxiong65190@163.com (B.X.); 2College of Integrated Circuits, Ludong University, Yantai 264025, China

**Keywords:** 3D trench electrode germanium detector, electrical characteristics simulation, heavy ions incidence, detector capacitance

## Abstract

High-purity germanium detectors, widely employed in fields such as aerospace applications based on radiation detection principles, have garnered attention due to their broad detection range and fast response time. However, these detectors often require larger sensitive area volumes to achieve larger signals and higher detection efficiency. Additionally, the large distance between the electrodes contributes to an issue of incomplete charge collection, which significantly restricts their application in space applications. To enhance the electrical performance of high-purity germanium detectors, this study introduces a strategy: designing the detector’s cathode electrode into a 3D trench shape with nested complementary cathodes. This design greatly reduces the electrode spacing, endowing the detector with superior electrical characteristics, such as a smaller dead zone and improved charge collection efficiency. Performance simulations of the novel detector structure were conducted using the semiconductor device simulation software Sentaurus TCAD (2019.03). The simulation results confirmed that the nested complementary 3D trench electrode high-purity germanium detector exhibits excellent electrical features, including a larger sensitive area volume, rapid charge collection, and good cell isolations. This approach has the potential to effectively expand the application scenarios of high-purity germanium detectors. Depending on different operational environments and requirements, nested complementary 3D trench electrode high-purity germanium detectors of appropriate structural dimensions can be chosen. The experimental findings of this study hold a significant reference value for enhancing the overall structure of high purity germanium detectors and facilitating their practical application in the future.

## 1. Introduction

Germanium, due to its characteristics such as a narrow bandgap, high electron mobility, and high energy resolution, has emerged as an ideal material to produce various types of detectors [1,2,3]. It has found widespread applications in fields such as nuclear physics, dark matter detection, and medical imaging [4,5,6,7,8]. High-purity germanium detectors, successfully developed in the 1970s, replaced traditional germanium lithium-drift detectors due to their simple fabrication processes and ease of room-temperature storage [9,10]. Several institutions are now capable of producing functional high-purity germanium detectors, which require a low-temperature operating environment to ensure low leakage current and high detection efficiency [11,12]. These detectors primarily come in two main types: planar and coaxial [13,14,15,16,17,18]. Both types require relatively thick sensitive volume areas for radiation detection, which can lead to a decrease in the ability to capture electron-hole pairs. To enhance the charge collection performance of the detectors, studies reported in relevant literature suggest increasing the detector thickness, thus causing an increasing in the reverse voltage between the anode and cathode, while keeping the increase in reverse current small [19,20]. This approach has limitations in terms of improving the charge-capture capability due to possible non-depleted region. To address this, a novel high-purity germanium detector structure needs to be developed. This structure aims to maintain a larger detector thickness while shortening the distance between the anode and cathode, achieving an effective reduction in the operation bias while having an increased charge collection efficiency of the high-purity germanium detector. Simultaneously, the overall detector performance can be greatly improved.

The structure of 3D trench electrode semiconductor detectors offers a new approach to enhancing the performance of high-purity germanium detectors [21,22]. This type of detector features a high-resistivity silicon detection substrate with a centrally positioned collection anode surrounded by a cathode ring with peripheral trenches. The distance between the anode and cathode determines the detector full depletion and it is independent of the wafer thickness. The detector cells in a pixel detector are relatively independent, exhibiting a more uniform internal electric field distribution, rapid charge collection speed, and short response time. Consequently, this significantly improves the detector’s radiation tolerance and charge collection efficiencies. Furthermore, due to the small area of the collection anode, the detector has a low capacitance, which is advantageous for reducing noise and enhancing energy resolution.

However, 3D trench electrode silicon detectors have some limitations. First, due to the inherent characteristics of silicon material, Si detectors have a small energy detection range (≤30 keV). While complex high-dimensional radiation environments require higher demands on the detector’s radiation detection range. In comparison, germanium material, with its larger atomic number and narrower bandgap, holds an advantage in the detection range and signal size. Incorporating germanium material into the field of 3D trench electrode semiconductor detectors is an effective approach to expanding the scope of semiconductor detector applications. Second, in the design of the 3D trench electrode detector structure, the cathode surrounding the perimeter trenches cannot penetrate the substrate, resulting in a dead region at the bottom of the detector. Existing literature has reported solutions to this issue, including adding cathode ring structures at the bottom and perforating the perimeter cathode [23]. While experimental and simulation results have met expectations, the intricacies of these structures hinder large-scale production, because the current technology is unable to produce detector chips with such intricate features.

Developing a detector with a smaller dead region while ensuring its exceptional performance and effectively minimizing the impact between adjacent units in the detector array has become a crucial issue in the field of radiation detection. High-purity germanium detectors, known for their wide detection range, coupled with the 3D trench electrode structure, offer the potential to significantly shorten the electrode spacing. However, introducing the 3D trench electrode structure into high-purity germanium detectors presents challenges: Firstly, unlike silicon, germanium cannot naturally form an oxide layer like Si dose in a thermal oxidation process, necessitating the identification of a suitable passivation layer material. Secondly, finding a way to reduce the dead region area of the detector while maintaining overall performance excellence. Thirdly, since a detector array consists of multiple detector units, mutual interference occurs during the detection signal process among these units. Inter-unit interference in detector arrays is one source of device noise, and the magnitude of the interfering signal directly impacts the detector’s performance. Therefore, minimizing the coherence between detector units is essential.

Building upon existing literature, this article proposes a nested 3D trench electrode high-purity germanium detector array that meets the requirements of a small dead zone and minimal interference between adjacent units. The proposal suggests using high-purity germanium material, which possesses a broad energy detection range, as the base material for the 3D trench electrode detector array. This material is employed to design a 3D trench electrode high-purity germanium detector with a nested complementary cathode structure. As high-purity germanium lacks a natural passivation layer, atomic layer deposition technology is used to coat aluminum oxide on the surface of the high-purity germanium detector as a passivation layer. To address the dead region issue at the bottom of the 3D trench electrode germanium detector, a complementary nested cathode structure is added around the central anode of the detector structure’s base.

The design of a 3D trench electrode germanium detector with nested complementary cathodes demonstrates a high level of feasibility. The process flow of a coaxial high purity germanium detector with a P-type substrate is taken as an illustrative example [24]. The raw material for the high purity germanium detector consists of a single crystal with a concentration of approximately 10^10^ cm^−^^3^. Following mechanical cutting, the high purity germanium wafer is formed. Subsequently, the n^+^ electrode layer is created through lithium diffusion on the outer surface of the detector, which undergoes grinding and chemical polishing. Finally, the p^+^ electrode is established via boron ion implantation to yield a coaxial high purity germanium detector. According to the relevant literature on the fabrication of 3D silicon detectors [22,25], we suggest that the realization of incorporating a 3D trench electrode structure into high purity germanium detectors necessitates several key technologies, including Deep Reactive Ion Etching, Thermal Diffusion, and Chemical Vapor Deposition. The Deep Reactive Ion Etching, and Thermal Diffusion technology are employed for fabricating the central n^+^ anode as well as the inner and outer p^+^ cathodes. Furthermore, a passivation layer of Al_2_O_3_ is deposited onto the high purity germanium substrate through Atomic Layer Deposition (ALD). With this ALD layer, Ge wafers can process similar to Si with SiO_2_ [26].

The 3D trench electrode germanium detector with nested complementary cathodes proposed in this paper has significant potential application value in pulsar navigation. Pulsar navigation employs X-ray detectors installed on a spacecraft to detect X-ray photons emitted by pulsar radiation, measure pulse arrival time, and determine navigation parameters such as spacecraft orbit, time, and attitude through processing corresponding signals and data [27]. Pulsars emit signals across various frequency bands ranging from radio waves, infrared light, visible light, ultraviolet rays to X-rays, and gamma rays. Although over 1000 pulsars have been discovered so far, the current silicon-based detectors are only suitable for detecting pulsar signals within the soft X-ray range, which severely limits the available options of pulsars for study. To address this limitation effectively, the development of a 3D trench high purity germanium detector is expected to yield positive outcomes. In comparison to silicon materials, germanium possesses a larger atomic number and a narrower bandgap width while exhibiting significantly higher carrier mobility than silicon does; these characteristics enable germanium detectors to operate at higher refresh rates in X-ray spectroscopy applications. Moreover, the discussed three-dimensional trench electrode high purity germanium detector is much smaller than traditional high purity germanium detectors while being capable of detecting radiation energy within both hard X-ray and low-energy gamma ray ranges.

Using the semiconductor device simulation software Sentaurus TCAD, this study conducted simulations of the electrical performance of the novel 3D trench electrode high-purity germanium detector. Starting from a single structure, the research extended to study detector arrays ranging from 3 × 3 to n × n. This resulted in 2D curves illustrating the electric field, potential, carrier concentration, and induced current over time. The study also investigated the coherence between adjacent detection units, revealing that a smaller distance between the nested bottom structure and the peripheral cathode results in reduced interference between neighboring units, thus strengthening isolation. In the final section of this paper, we conduct a simulation to evaluate the impact of detector capacitance. The magnitude of this capacitance directly influences both crosstalk noise and detector performance.

## 2. Detector Structure

The device was simulated using Sentaurus TCAD software. In this study, the Carrier Transport model, Shockley–Read–Hall (SRH) recombination model, and state density model were employed for analysis. The material parameters are directly obtained from the built-in material parameter database of Sentaurus TCAD software. We consider impurity scattering-induced carrier mobility degradation (DopingDep) using the ion mobility model, high electric field-induced carrier velocity saturation (HighFieldSat), and surface roughness scattering-induced mobility degradation (Enormal). The state density model selects the effective intrinsic density (OldSlotboom) due to the presence of doping. Additionally, the Recombination model activates corresponding carrier production-recombination phases in the carrier continuity equation and incorporates SRH recombination. Additionally, the interface charge density at the Al_2_O_3_-Ge interface was set to −4 × 10^11^ q/cm^2^.

As shown in Figure 1a, the schematic diagram illustrates the overall structure of the nested complementary 3D trench electrode high-purity germanium detector. The device has a prismatic shape, with dimensions of 100 μm × 100 μm × 302 μm, where the detector substrate thickness is 300 μm. The surface layer of the detector is 1 μm thick, and the cathode and anode contact materials are Al, with widths of 10 μm. Al_2_O_3_ is chosen as the passivation layer between the electrodes, serving as insulation and isolation. The base material of the detector is lightly doped N-type high-purity germanium with a doping concentration of 1 × 10^12^/cm^3^. The central anode size is 10 μm × 10 μm × 300 μm, spanning the detector substrate, with N-type heavily doped germanium with a doping concentration of 1 × 10^19^/cm^3^. The peripheral trench cathode that surrounds the central anode is located at the outermost part of the device unit cell, with a width of 10 μm and a thickness of 280 μm. It is P-type heavily doped germanium with a doping concentration of 1 × 10^19^/cm^3^. At the bottom of the device, Al_2_O_3_ is selected as an insulating protective layer with a thickness of 1 μm.

To address the issue of a dead region at the bottom of the detector, a nested complementary structure for the cathode at the bottom of the device is designed from the bottom up. After forming the detector array, each central anode needs to be surrounded by four bottom cathode walls, allowing the induced electrons to be collected and detected by the central anode. To further maximize the sensitive volume of the detector, as shown in Figure 1b(i)–(iii), three complementary structures for the bottom nested cathode are designed, differing only in the number of openings (i. fully enclosed, ii. single opening, iii. double opening). The doping concentration of these nested cathodes is consistent with that of the outer trench cathodes, both being 1 × 10^19^/cm^3^. The nested cathode has a width of 10 μm and a thickness of 40 μm. The gap between the nested cathode and the peripheral groove electrode, referred to as g, was set at 10 μm.

As shown in Figure 1b(iv), a single unit of the detector structure cannot meet the requirements of practical applications. Therefore, it is necessary to consider the study of detector arrays, selecting detector arrays of 3 × 3 up to n × n for structural design. From the figure, it can be observed that during the formation of the detector array, except for the edges of the detector, unit (iii) is a repetitive unit. Based on the differences in structural dimensions, a programmable design of the arrangement of the three unit types in the detector array is achieved.

## 3. Electrical Characteristics

The electrical simulation images of the nested complementary 3D trench electrode high-purity germanium detector with 3 × 3 array are presented in Figure 2. A reverse bias of −4 V is applied between the cathode and anode of the detector. The electron depletion zone inside the detector gradually extends toward the anode with voltage. When the voltage reaches the depletion voltage, the electron depletion zone reaches the anode, and the detector is fully depleted. As shown in Figure 2a, cross-sections are taken at z = 140 μm, y = 270 μm, and y = 290 μm, corresponding to the sections shown in Figure 2b, 2c, and 2d, respectively. (i) represents the doping distribution state of the detector, (ii) depicts the electric field distribution, (iii) shows the electrostatic potential distribution, and (iv) illustrates the electron concentration distribution.

The electric field distribution plot (ii) of the device reveals a uniform distribution within the substrate, indicating a uniform electric field to drive a precise electron transportation to the central anode. The potential distribution plot (iii) demonstrates that the central anode exhibits the highest potential, while the cathode displays the lowest potential. This configuration effectively facilitates electron transport towards the central anode, thereby accomplishing rapid detector response due to small electrode spacing and uniform electric field. In the electron concentration distribution plot (iv), the electron concentration exhibits its highest values near the anode, while most regions within the substrate are in a depleted state.

Planar high purity germanium detector with thickness of 20 mm and full depletion voltage of −1300 V has been reported in the literature [28]. The full depletion voltage of the nested complementary 3D trench electrode high purity germanium detector studied in this paper is −4 V, even if its structure is made into a thickness of 20 mm, the depletion voltage is only required to be −4 V, because the full depletion voltage of the detector structure designed in this paper is independent of the wafer thickness.

The simulation results demonstrate a significant reduction in the dead region of the bottom detector, while maintaining excellent overall detector performance. However, due to the incomplete closure of the bottom nested structure, potential mutual interference in charge collection between adjacent units still persists. Although this interference is small, it can slightly impact the detection performance of the detector.

As depicted in Figure 3, this study selected two adjacent repetitive units (Figure 1b(iv) within the detector array for investigation. A beam of heavy ions was incident at the middle position between the cathode and anode, located inside unit ‘a’ along the vertical direction of the substrate. The initial time of incidence was set to 5 ns, with an incidence depth of 100 μm, action range radius set at 0.2 μm, and linear energy conversion rate at 1 × 10^−^^5^ pC/μm.

Due to the incidence of heavy ions, the initially fully depleted detector experienced the generation of electron-hole pairs. Subsequently, under reverse bias conditions, these electron-hole pairs drift towards the anode and cathode, respectively, generating an electrical signal.

The changes in electron concentration within the detector before and after heavy ion incidence are illustrated in Figure 3i–vii. Initially, at 3 ns (i), the detector was fully depleted. Subsequently, at 5 ns (ii), heavy ions strike the substrate of the detector, resulting in the formation of a high-density electron region within it. This region gradually migrated towards the central anode over time until it was completely absorbed by the anode (iii–vii).

Figure 4a depicted the time-varying curve of the central anode current of unit ‘a’. Following the incidence of heavy ions on unit ‘a’ of the detector at 5 ns, the induced current of unit ‘a’ exhibited a rapid increase until reaching its peak, followed by a subsequent decrease. Once all electrons were completely collected by the central anode, the anode current returned to its depleted state.

Meanwhile, the incidence of heavy ions occurring in unit ‘a’ would have an impact on adjacent unit ‘b’. As shown in Figure 4b, the time-varying curve of the central anode current in unit ‘b’ is presented. The curve indicates that the heavy ions incident in unit ‘a’ indeed affected adjacent unit ‘b’. Over time, unit ‘b’ experienced an increase in current followed by a decline to the depleted state, with a peak of 5.07 nA, 65 times smaller than the peak of 330 nA in unit ‘a’. To further mitigate mutual interference between detector units and ensure maximum independence of detector units, this study investigated the coherence of detector units by varying the distance ‘g’ between the bottom cathode nested structure and the peripheral cathode structure.

In the simulations, ‘g’ was set at intervals of 10, 8, 6, 4, and 2. Considering the necessity of accounting for interconnection between substrates in actual production to prevent electrode detachment, ‘g’ was ultimately chosen as 2 μm instead of 0 μm (which would directly connect the bottom cathode nested structure with the peripheral cathode structure).

As depicted in Figure 4, among the five control simulations, there were no significant differences observed in the time-varying current collected by unit ‘a’ (Figure 4a). The anode currents of unit ‘b’ all peaked simultaneously and subsequently returned to a depleted state. Notably, at g = 10 μm, the interference current exhibited its highest magnitude, gradually decreasing as ‘g’ decreased and reaching its minimum at g = 2 μm (Figure 4b). This suggests that reducing the value of ‘g’ can enhance the isolation of detector units, provided that the process technology meets the required standards.

The simulation results demonstrate that the controllability of detector performance is influenced by the distance ‘g’ between the inner and outer cathodes. It is crucial to select a detector structure suitable for actual production under specific equipment and production technology conditions. The Deep Reactive Ion Etching technique imposes limitations on achieving an optimal g value. If g is too small, it not only increases technical difficulties but also promotes adhesion between internal and external nested cathodes, compromising the effectiveness of the g value. Moreover, a decrease in mechanical strength may lead to detachment of the detector structure from its substrate.

From the graph, it becomes evident that for g = 10 μm, the nested complementary 3D trench electrode detector exhibits the highest interference current in adjacent unit ‘b’. As the value of ‘g’ decreases, the interference current exhibits a small reduction, reaching its minimum at g = 2 μm. However, since interference at g = 10 μm is already very small, it may be considered insignificant and a g = 10 μm may be good enough for most applications.

The crosstalk noise significantly impacts the performance of the radiation detector. The magnitude of crosstalk noise in the detector is contingent upon its capacitance characteristics. To investigate these characteristics, we selected the 3 × 3 units array depicted in Figure 2a as our research subject and simulated the capacitance between the cathode and anode of its central unit (the repeating unit of the array). A reverse bias was applied across these electrodes, with a voltage of 0 V at the cathode and −100 V at the anode.

As depicted in Figure 5, the capacitance of the detector unit exhibits variations under reverse bias conditions. The increase in depletion region thickness leads to a decrease in capacitance. At full depletion state, the depletion capacitance reaches its minimum value. For a reverse bias voltage of 100 V, the corresponding capacitors for values 2, 4, 6, 8, and 10 are measured as 2.52 pF, 0.39 pF, 0.15 pF, 2.12 pF, and 5.82 pF, respectively.

When g ranges from 2 to 6 inclusive, there is a decreasing trend observed in capacitance; however, when g ranges from 6 to 10 inclusive, an upward trend is observed instead. We attribute this change in trend to the nested cathode structure at the bottom of the detector unit design where each central anode requires four surrounding bottom cathode walls for efficient charge signal collection (Figure 1b). During formation of the detector unit array configuration adjacent units share common bottom nested cathodes which results in increased distance between anodes and cathodes within individual units while reducing distances between adjacent units simultaneously. Therefore, it becomes necessary to consider the impact of this bottom nested structure on detector capacitance during selection of the appropriate detector configurations.

## 4. Conclusions

To address the issue of incomplete charge collection in high-purity germanium detectors due to excessive electrode spacing and to further enhance detector efficiency, this paper proposes a 3D trench electrode high-purity germanium detector with nested complementary cathode structure. The cathode structure of this type of detector is designed in a trench shape, and the nested complementary cathode structure is added to the bottom of the detector to eliminate the detection dead region. The novel detector structure was electrically simulated using the semiconductor simulation software Sentaurus TCAD, and the coherence between the adjacent detector units was investigated. The simulation results show that the distance between the detector’s cathode and anode has significantly decreased, resulting in excellent detection performance. Additionally, it exhibits good unit independence, and the distance between the bottom nested structure and the peripheral cathode can be adjusted to meet different testing requirements. Meanwhile, even at g = 10 μm, this inter-cell interference may be insignificant. In the final section of this paper, we investigate the correlation between detector capacitance and bias voltage. The findings demonstrate that the detector capacitance is contingent upon the configuration of the underlying nested cathode and is influenced by the shared bottom cathode wall among adjacent detection units. Consequently, it holds immense significance to carefully select an appropriate g value in order to regulate detector capacitance effectively and thereby control crosstalk noise.

## Figures and Tables

**Figure 1 micromachines-14-02051-f001:**
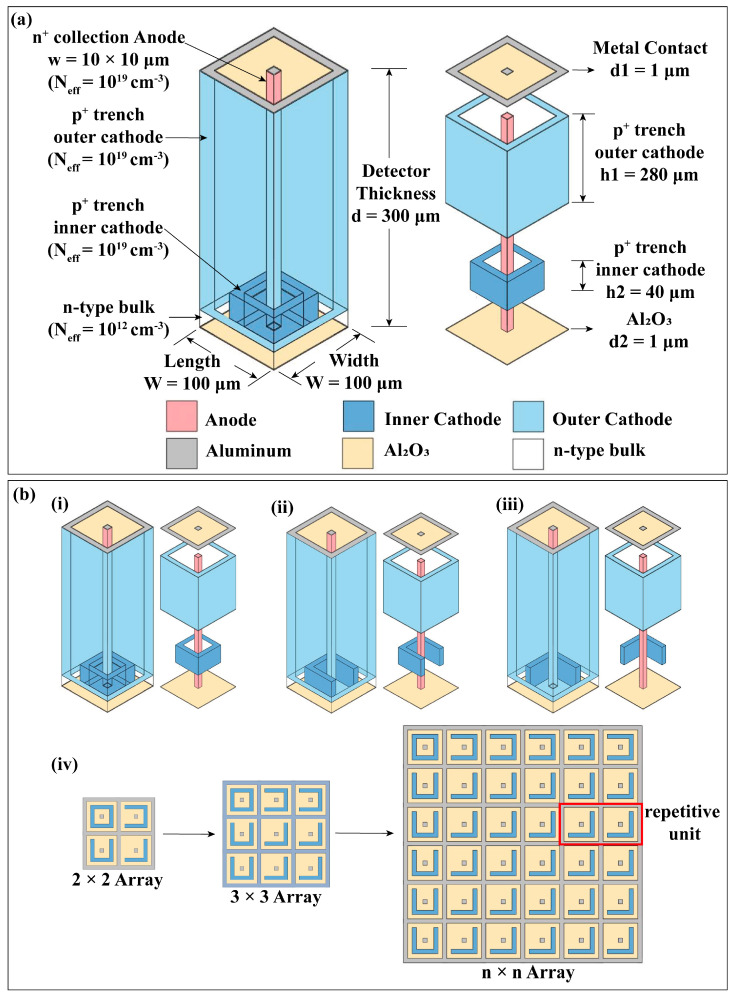
Schematic diagram of the 3D trench electrode germanium detector with nested complementary cathodes. (**a**) Overall detector structure (left) and exploded view (right). (**b**) Detector array formed by the combination of three different unit structures with nested complementary cathodes. The distinction between unit structures (**i**–**iii**) lies solely in the number of openings in the bottom nested cathode walls. (**i**) Fully enclosed, (**ii**) single opening, and (**iii**) double openings. (**iv**) Following the principle of surrounding the central anode with four nested cathode walls, appropriate combinations of the three units yield 2 × 2, 3 × 3, n × n arrays. The repeated unit (**iii**) in the array is highlighted by a red outline.

**Figure 2 micromachines-14-02051-f002:**
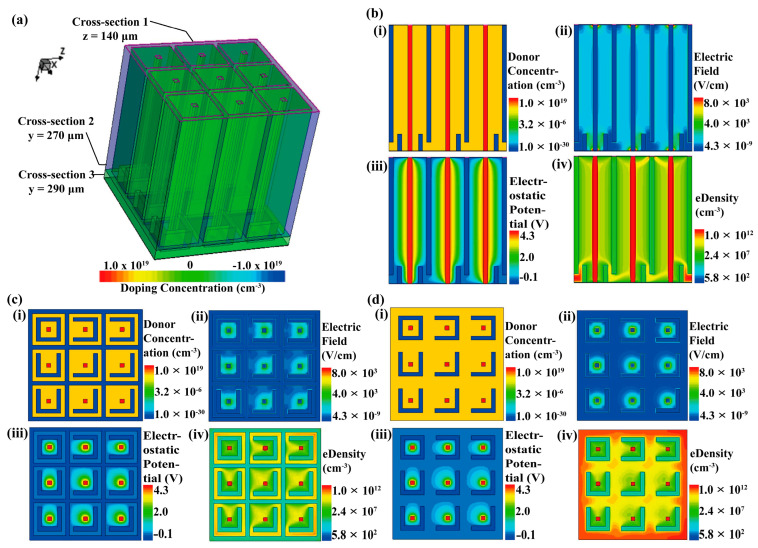
Electrical simulation images of a 3 × 3 array of nested complementary cathode 3D trench electrode high-purity germanium detectors. (**a**) 3D structural simulation images. Cross-section 1: Vertical cut along the midpoint of the array width (z = 140 μm). Cross-section 2: Horizontal cut along the array length (y = 270 μm). Cross-section 3: Horizontal cut along the array length (y = 290 μm). (**b**) Electrical simulation results at the cross-section 1. (**i**) Donor concentration; (**ii**) Electric field; (**iii**) Electric potential; (**iv**) Electron concentration. (**c**) Electrical simulation results at the cross-section 2. (**i**) Donor concentration; (**ii**) Electric field; (**iii**) Electric potential; (**iv**) Electron concentration. (**d**) Electrical simulation results at the cross-section 3. (**i**) Donor concentration; (**ii**) Electric field; (**iii**) Electric potential; (**iv**) Electron concentration.

**Figure 3 micromachines-14-02051-f003:**
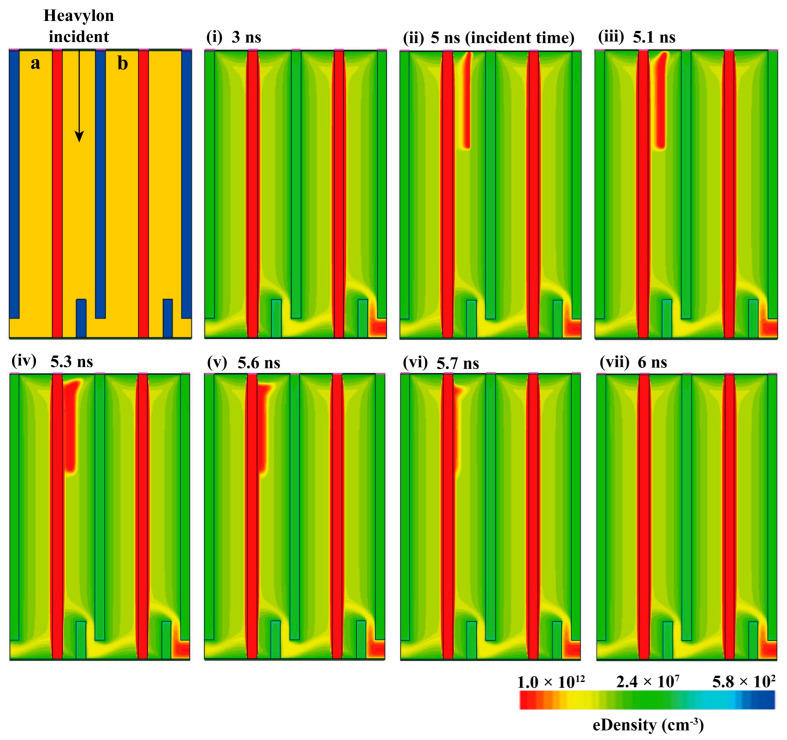
Changes in electron concentration within the detector before and after heavy ion incidence. Left 1: ‘a’ and ‘b’ represent two adjacent units, with a heavy ion beam vertically irradiating the middle position between the cathode and anode of unit ‘a’. (**i–vii**) Images depicting the temporal variation of electron concentration inside the detector before and after heavy ion incidence. (**i**) 3 ns; (**ii**) 5 ns, time of heavy ion incident; (**iii**) 5.1 ns; (**iv**) 5.3 ns; (**v**) 5.6 ns; (**vi**) 5.7 ns; (**vii**) 6 ns.

**Figure 4 micromachines-14-02051-f004:**
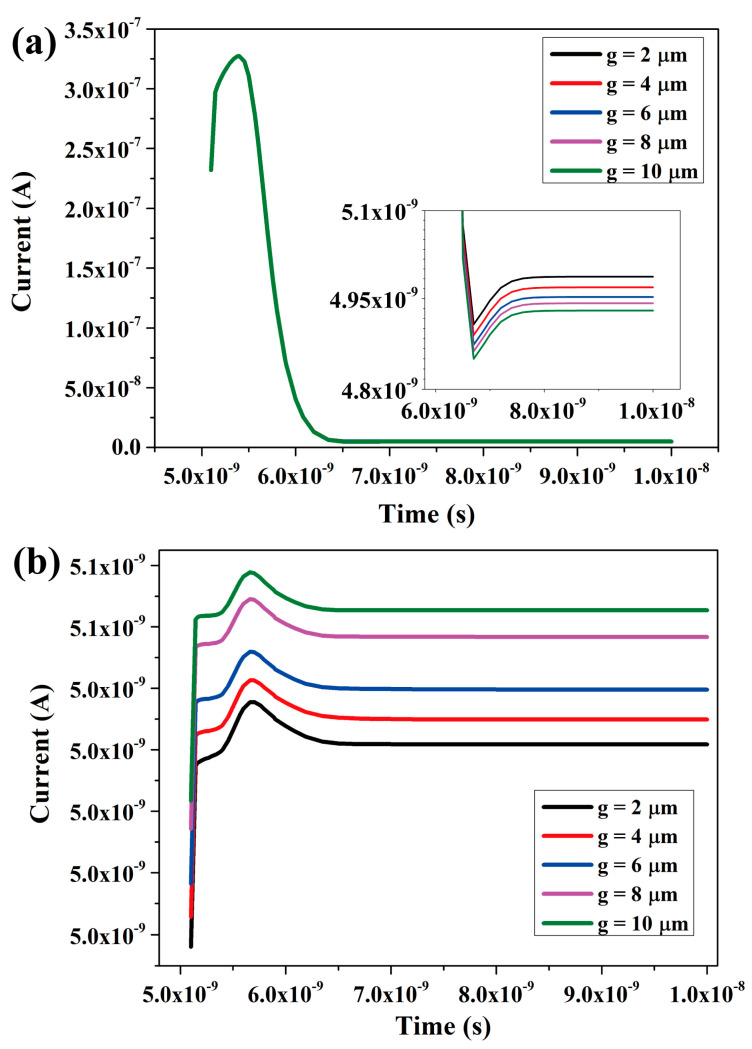
When heavy ions are incident into unit ‘a’, detectors with different cathode spacings ‘g’ show variations in the electron current collected by the anode over time. (**a**) Time-varying curve of the electron current collected by the anode of unit ‘a’. (**b**) Time-dependent curve of the electron current collected by the anode of unit ‘b’.

**Figure 5 micromachines-14-02051-f005:**
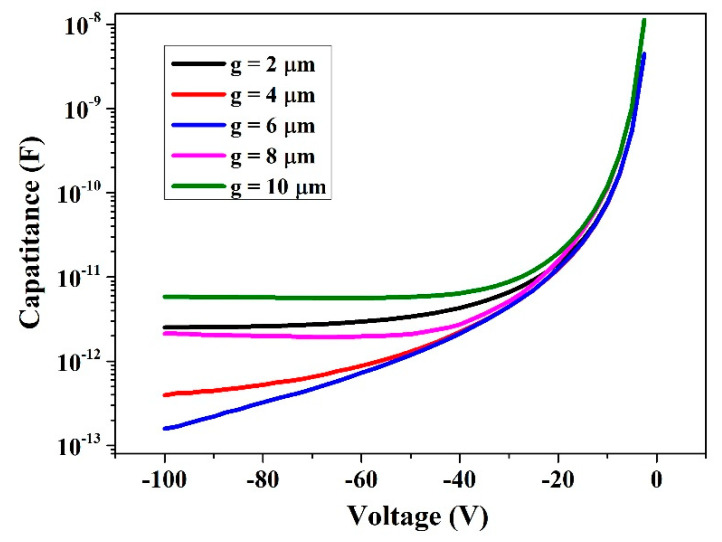
The variation of detector capacitance under reverse bias conditions.

## Data Availability

The data presented in this study are available from the corresponding author upon reasonable request.

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
