# Peer review of "Electrical Characteristics of 3D Trench Electrode Germanium Detector with Nested Complementary Cathodes"

_micromachines, 2023, doi:10.3390/mi14112051_

Round 1

Reviewer 1 Report

Comments and Suggestions for Authors

Dear authors,

The premise of this manuscript, using the 3D electrode approach on germanium detectors to enhance their electrical performance, starts very interesting and innovative but does not reflect fully the expected content described within the abstract.

I highly recommend to go more into detail in this topic considering how impactful such a detector technology might be in future, especially in X-ray spectroscopy, allowing to operate the detector at higher refresh rates.

The last sentence from the abstract "The experimental outcomes of this study hold significant guiding implications for improving the structure of high-purity germanium detectors and facilitating mass production." results a bit miss- orientating to the reader, who is expecting some indications on how to produce this kind of device. If this was the author's intent, I suggest to perform some sentaurus sprocess simulations in which you can test the processes necessary to realise the device. Otherwise, I would reformulate this statement.

Related to the last quote, the electrode structure should be designed closer to a more feasible design. A more simplified design of the sensor is also recommended, by applying today-used electrode geometries. 

line 53-54: "The concept of 3D trench electrode detectors was first introduced by the Brookhaven National Laboratory in the United States[22,23]" In fact 3D electrode sensors with trench electrode were firstly introduce in 2011 by a collaboration of University of Hawaii, SINTEF, SLAC, CERN, EPFL and Univeristy of Manchester (S. Parker et al., "Increased Speed: 3D Silicon Sensors; Fast Current Amplifiers," in IEEE Transactions on Nuclear Science, vol. 58, no. 2, pp. 404-417, April 2011, doi: 10.1109/TNS.2011.2105889.)

Reviewer 2 Report

Comments and Suggestions for Authors

The paper proposes an interesting study on electrical characteristics of 3D trench electrode Germanium detectors, but some aspects must be improved, in my opinion, before publication. I hope the comments below will help to improve the article.

In the introduction you should add some information on the germanium detectors  manufactured in literature, in particular related to the possibility to realize 3D germanium detectors. You should add information about the process to realize 3D detectors in germanium, adding more literature details about the possibility to manufacture p+ trench outer cathode and inner cathode.

In the introduction you should add some information about the application of the 3D germanium detectors. Add please the specific field in which you want to use these detectors.

In the section 2 “Detector structure” you should add more details on the model you have used in the TCAD to simulate the device. You should add information about the germanium parameters you have used in the TCAD and add the respective references.

You should add information on the surface model you have used. How have you simulated the interface between the passivation and the germanium. Add literature references related to this topic.

You should add information about the physics models (mobility, recombination ecc. Ecc.) of the TCAD model you have used.

In section 3 you should show a comparison between simulated and experimental measurements on a simple planar device demonstrating the validity of the modelling scheme used, before to use the model to analyze different layouts of 3D devices. If you have not the possibility to characterize a simple planar device, you can compare the simulated data with experimental ones extracted from literature.

You should add more information about the “heavy ion” model you have used (rows 189-193) adding the literature references you have considered for the electron-hole couples generation in the germanium.

You have varied the parameter g in the range 0.01- 10 micron. Which is the technological limit for this parameter? In my opinion you should describe the actual possibilities for this parameter, exploring the advantages of using this parameter inside the actual technological limits and then you can explore one value beyond limits.

You should analyze the capacitance characteristics of your device, because the capacitance is a very important parameter. You should analyze the capacitance characteristics behavior as a function of the parameter “g”.

Round 2

Reviewer 1 Report

Comments and Suggestions for Authors

Dear authors,

Thanks for having updated and corrected the content.

Reviewer 2 Report

Comments and Suggestions for Authors

The paper has been improved taking into accounts the comments. I have only one minor revision.

The question is related to the depletion voltage. In rows 231-236 you write that the depletion voltage is -4 V, while in fig. 5 you show that the depletion voltage is higher for all the cases you have considered. 

Comments on the Quality of English Language

I have a comment related to this sentence: 

The material parameters utilized are directly obtained from the built-in material parameter database of Sentaurus TCAD software. Considering impurity scattering-induced carrier mobility degradation (DopingDep) through the ion mobility model, high electric field-induced carrier velocity saturation (HighFieldSat), and surface roughness scattering-induced mobility degradation (Enormal).

Probably you have to rewrite adding a common before "Considering...."